# AFFINE INVARIANCE IN CONTINUOUS-DOMAIN CONVOLUTIONAL NEURAL NETWORKS

## ABSTRACT

The notion of group invariance helps neural networks in recognizing patterns and features under geometric transformations. Indeed, it has been shown that group invariance can largely improve deep learning performances in practice, where such transformations are very common. This research studies affine invariance on continuous-domain convolutional neural networks. Despite other research considering isometric invariance or similarity invariance, we focus on the full structure of affine transforms generated by the generalized linear group $\mathrm{GL}_2(\mathbb{R})$. We introduce a new criterion to assess the similarity of two input signals under affine transformations. Then, unlike conventional methods that involve solving complex optimization problems on the Lie group $G_2$, we analyze the convolution of lifted signals and compute the corresponding integration over $G_2$. In sum, our research could eventually extend the scope of geometrical transformations that practical deep-learning pipelines can handle.

## 1 INTRODUCTION

Convolutional neural networks (CNNs) have achieved remarkable success at analyzing, recognizing, and understanding of images. The capability of these networks is largely attributed to their ability to generate good features from raw data. However, the types of structures that CNNs can explore are limited to simple symmetries.

Addressing this limitation, Group Convolutional Neural Networks (G-CNNs) extend CNNs to capitalize on the intrinsic geometric properties and symmetries in data, particularly images (Cohen & Welling, 2016). Unlike their traditional counterparts, G-CNNs harness the power of group theory, a mathematical framework that formalizes transformations and symmetries. This theoretical foundation ensures equivariance with respect to transformations described by the group, thereby enabling the network to maintain predictable behavior under various transformations.

One striking characteristic of G-CNNs is their ability to preserve and leverage the fundamental structures of data throughout the network's architecture. Notably, they excel when dealing with large groups that extend beyond mere translation equivariance. Classical CNNs can be regarded as a special instance of G-CNNs. The real power of G-CNNs becomes evident when more intricate geometric transformations are at play. Recent G-CNNs elevate feature maps to higher-dimensional, disentangled representations (Bekkers, 2019). Within these representations, G-CNNs effectively learn the characteristics of the data, rendering traditional geometric data-augmentation techniques superfluous. This not only streamlines the learning process but also minimizes the risk of overfitting. Moreover, G-CNNs maintain their predictive behavior under geometric transformations, thanks to their foundation in group theory and, therefore, give rise to the concept of equivariance. The introduction of G-CNNs to the machine-learning community by Cohen & Welling (2016) marked the inception of an expanding body of G-CNN literature that consistently highlights many advantages of G-CNNs over conventional CNNs. This literature can be roughly classified into three main categories: discrete G-CNNs, regular continuous G-CNNs, and steerable continuous G-CNNs. Discrete G-CNNs delve into discrete group structures, yielding improved performance in various applications. This approach has been explored in studies by Cohen & Welling (2016); Winkels & Cohen (2018); Dieleman et al. (2016); Worrall & Brostow (2018); Hoogeboom et al. (2018), collectively contributing to the foundational understanding and practical deployment of discrete G-CNNs.

Regular continuous G-CNNs, as investigated by Oyallon & Mallat (2015); Bekkers et al. (2015); Weiler et al. (2018); Zhou et al. (2017), focus on seamless transformations within continuous domains. Their research showcases how G-CNNs can excel in handling continuous data, offering advantages over traditional CNNs in capturing intricate patterns and representations. Steerable continuous G-CNNs, explored Cohen et al. (2018); Worrall et al. (2017); Kondor & Trivedi (2018); Thomas et al. (2018); Andrearczyk et al. (2019), introduce a specialized approach where the convolution kernels are represented in terms of circular or spherical harmonics. This technique, particularly suitable for unimodular groups like roto-translations, enables efficient computation by utilizing basis coefficients.

Our research investigates the property of affine invariance in the context of continuous-domain convolutional neural networks. Our focus are affine spaces formed by the generalized linear group $\mathrm{GL}_2(\mathbb{R})$, the group of all invertible matrices of size $2 \times 2$. Affine transformations are fundamental operations that combine linear transformations and translations. These transformations are important because they address distortions of an affine nature. For example, such distortions arise in photos when the camera is close to the subject being captured Fisher et al. (2000); Guo et al. (2019), or in certain types of CAPTCHA Wang & Lu (2018) (see Figure 1). Previous attempts have been made to investigate spaces that maintain affine-equivariance, but they are restricted to strict conditions, such as cases where the determinant equals 1 (expressed as $\mathrm{SO}(n)$). We instead consider affine-invariant spaces across the entire spectrum of invertible matrices. The conventional method for determining the invariance of two input signals, denoted as $f_1$ and $f_2$, under a transformation involves solving a complex optimization problem on the Lie group $G_2$. Instead, we introduce an alternative approach, where we assess whether the convolution of the lifting of $f_1$ and $f_2$ to $G_2$ exhibits $G_2$ invariance for every kernel. In order to apply this criteria, an additional step is required, namely, the computation of convolutions over $G_2$. We Solve this technical challenge using QR-decomposition discussed in Schindler (1993).

Two main contributions of this paper are:

- Rather than focusing on complex optimization problems, we study invariance through convolution integrals in the group space.
- We show how to perform the related computations by simplifying the convolutions over the transformation group to integrals over real space.

In this way, we are able to cater to a considerably broader spectrum of transformations. This result is very broad and, in specific scenarios, it can be used to analyze invariance in affinely generated transformations, such as the Roto-translation transformation.

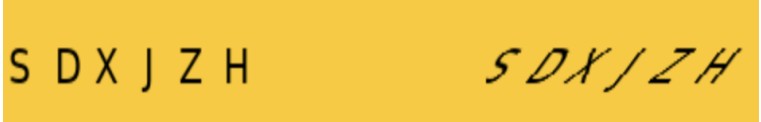

Figure 1: Original letters and its affine invariant CAPTCHA.

## 1.1 PRELIMINARIES

This section establishes the foundational concepts, terminology, and context along with some illustrative examples. These preliminaries set the stage for the main contributions and discussions presented in the following sections.

Ensuring the equivariance of artificial neural networks (NNs) with respect to a group $G$ is an essential characteristic, as it guarantees that applying transformations to the input preserves all information, merely shifting it to different network locations. It has been determined that when aiming for equivariant NNs, the sole viable choice is to employ layers in which the linear operator is defined through group convolutions. The journey to this conclusion commences with the conventional definition of neural networks layers given by

$$\boldsymbol{y} = \sigma\left(\mathcal{K}_{\boldsymbol{w}} f(\boldsymbol{x}) + \boldsymbol{b}\right), \tag{1}$$

where $\boldsymbol{x} \in \mathcal{X}$ represents the input vector, and $f$ denotes the input signal, for example in an image input, $\boldsymbol{x}$ is the location of pixels and $f$ is the value that image takes in each pixel. Moreover, $\mathcal{K}_{\boldsymbol{w}} : \mathcal{X} \to \mathcal{Y}$ stands as a linear map parameterized by the weight vector $\boldsymbol{w}$, with $\boldsymbol{b} \in \mathcal{Y}$ as a bias term, and $\sigma$ is the activation function. The kernel operator $\mathcal{K}_{\boldsymbol{w}}$ is also defined as follows

$$\mathcal{K}_{\boldsymbol{w}} f = \int_{\mathcal{X}} k(\boldsymbol{x}, \boldsymbol{y}) f(\boldsymbol{x}) d\mu_{\mathcal{X}}(\boldsymbol{x}),$$

where $\mathcal{K}_{\boldsymbol{w}} : \mathbb{L}_2(\mathcal{X}) \to \mathbb{L}_2(\mathcal{Y})$, $d\mu_{\mathcal{X}}$ is a Radon measure on $\mathcal{X}$, $k$ denotes the kernel function, and $f \in \mathbb{L}_2(\mathcal{X})$ (square integrable function). To broaden the application of this explanation to the notion of group convolutional neural networks, we revisit a number of crucial definitions.

**Definition 1** (Group). *A group $(G, \cdot)$ is a set $G$ equipped with a binary operator represented by a dot symbol. The dot operator is associative $((g_1 \cdot g_2) \cdot g_3 = g_1 \cdot (g_2 \cdot g_3))$, has an identity element $(e)$. Moreover, every element of the set has an inverse element $(g \cdot g^{-1} = g^{-1} \cdot g = e)$.*

In this context, the set comprises functions, such as translations or rotations. The group operation operates on elements of this set through addition or multiplication (Herstein, 1991). We also need to define normal groups. A normal group is a subgroup $N$ of a group $G$ such that, for every element $g$ in $G$, the conjugate $gNg^{-1}$ is contained within $N$.

**Example 1** (Translation group). *The translation group in $\mathbb{R}^2$ is denoted by $(\mathbb{R}^2, \cdot)$ consists of all possible translations and is equipped with the below group product and group inverse:*

$$g \cdot g' = (\boldsymbol{x} + \boldsymbol{x}')$$
$$g^{-1} = -\boldsymbol{x},$$

*where $g = (\boldsymbol{x})$ and $g^{-1} = (-\boldsymbol{x})$ and $\boldsymbol{x}, \boldsymbol{x}' \in \mathbb{R}^2$.*

One important example of groups are Lie groups, which are defined as follows:

**Definition 2** (Lie groups). *Special case of groups are Lie groups, which are symmetries of Riemannian manifolds.*

Roto-translation symmetries of Euclidean spaces are examples of Lie groups, which is explained in the next example.

**Example 2** (Roto-translation group). *The roto-translation group in $\mathbb{R}^2$ is denoted by $\mathrm{SE}(2)$. The group $\mathrm{SE}(2) = \mathbb{R}^2 \rtimes \mathrm{SO}(2)$ (where $\rtimes$ denotes semidirect product. In a direct product $G = H \times K$, both $H$ and $K$ are normal in $G$. Semidirect products are a relaxation of direct products where only one of the two subgroups must be normal) consists of translations vectors in $\mathbb{R}^2$, and rotations in $\mathrm{SO}(2)$ and is equipped with the group product and group inverse:*

$$g.g' = (\boldsymbol{x}, \boldsymbol{R}_\theta) \cdot (\boldsymbol{x}', \boldsymbol{R}_{\theta'}) = (\boldsymbol{R}_\theta \boldsymbol{x}' + \boldsymbol{x}, \boldsymbol{R}_{\theta+\theta'})$$
$$g^{-1} = (-\boldsymbol{R}_\theta^{-1} \boldsymbol{x}, \boldsymbol{R}_\theta^{-1}),$$

*for $g = (\boldsymbol{x}, \boldsymbol{R}_\theta)$, $g' = (\boldsymbol{x}', \boldsymbol{R}_{\theta'})$, and*

$$\boldsymbol{R}_\theta = \begin{pmatrix} \cos\theta & -\sin\theta \\ \sin\theta & \cos\theta \end{pmatrix}.$$

The group operator provides instructions on how to act on the group elements, ensuring that the result remains within the group. Of particular interest are symmetry groups, where each element in the set represents a symmetry transformation. When the group acts on a specific space, it is referred to as a group action.

**Definition 3** (Group action). *Let $\chi$ be a set. If $G$ is a group with identity element $e$, then a group action $\alpha$ of $G$ on $\chi$ is a function, $\alpha : G \times \chi \to \chi$, that satisfies identity and compatibility conditions $(e \odot \boldsymbol{x} = \boldsymbol{x}, g \odot (h \odot \boldsymbol{x}) = (g \cdot h) \odot \boldsymbol{x})$ for all $g, h \in G$ and all $\boldsymbol{x} \in \chi$.*

For example the action of group $G = \mathrm{SO}(d)$ on space $\chi = \mathbb{R}^d$ could be denoted by $g \odot \boldsymbol{x} = \boldsymbol{Rx}$, where $\boldsymbol{x} \in \mathbb{R}^d$ and $\boldsymbol{R} \in \mathrm{SO}(d)$. For the set of points, we perform transformation through group products. While in the convolution kernel, we perform transformation via group representations. Therefore we need to understand representations. The multiplication within a group instructs us on merging transformations, yet it does not provide guidance on utilizing these transformations on other entities like vectors or signals. To address this, we require the concept of group action and group representations. Nevertheless, frequently, our attention is predominantly directed towards linear group actions operating on vector spaces, and these actions are termed representations.

**Definition 4** (Representation). *A representation is an invertible linear transformation $\rho(g) : V \rightarrow V$ parameterized by group elements $g, h \in G$ that acts on some vector space $V$, which follows the group structure (it is a group homomorphism) via*

$$\rho(g)\rho(h)v = \rho(g \cdot h)v$$

*for $v \in V$.*

**Definition 5** (Regular representation). *Let $f \in \mathbb{L}_2(\mathcal{X})$. Then the regular representation of $G$ acting on $\mathbb{L}_2(\mathcal{X})$ is given by*

$$\rho(g)f(\boldsymbol{x}) = f\left(g^{-1}\boldsymbol{x}\right).$$

**Example 3** (Regular representation of roto-translation group). *Let $f \in \mathbb{L}_2(\mathbb{R}^2)$ be a two dimensional image, $G = \mathrm{SE}(2)$ denotes the roto-translation group then*

$$\rho(g)f(\boldsymbol{y}) = f(R_\theta^{-1}\boldsymbol{y} - \boldsymbol{x}).$$

We continue this part with some additional definitions that we need in the next section.

**Definition 6** (Coset). *Let $H \subset G$ be a subgroup of $G$. Then $gH$ denotes a coset given by*

$$gH = \left\{ g \cdot h \mid h \in H \right\}.$$

**Definition 7** (Quotient Space ). *Let $H \subset G$ be a subgroup of $G$. Then $G/H$ denotes the quotient space that is defined as the collection of unique cosets $gH \subset G$. Elements of $G/H$ are thus cosets that represents an equivalence class of transformations for which $g \sim \tilde{g}$ are equivalent if there exsists a $h \in H$ such that $g = \tilde{g}h$.*

**Definition 8** (Stabilizer). *Let $G$ acts on $\mathcal{X}$ via the action $\odot$. For every $\boldsymbol{x} \in \mathcal{X}$, the stabilizer subgroup of $G$ with respect to the point $\boldsymbol{x}$ is denoted with $\mathrm{Stab}_G(\boldsymbol{x})$ is the set of all elements in $G$ that fix $\boldsymbol{x}$*

$$\mathrm{Stab}_G(\boldsymbol{x}) = \left\{ g \in G \mid g \odot \boldsymbol{x} = \boldsymbol{x} \right\}.$$

Moreover from (Bekkers, 2019) we know that, if $\mathcal{X}$ be a homogeneous space of $G$. Then $\mathcal{X}$ can be identified with $G/H$ with $H = \mathrm{Stab}_G(\boldsymbol{x}_0)$ for any $\boldsymbol{x}_0 \in \mathcal{X}$. Finally we have $\epsilon$-Affine invariance definition.

**Definition 9** ($\epsilon$-Affine invariance). *We say that functions $f_1, f_2 \in \mathbb{L}(\mathbb{R}^2)$ are $\epsilon$-Affine invariant if there exists $\boldsymbol{A} \in G_2$ so that $\|f_1 - \rho(\boldsymbol{A})f_2\|_1 < \epsilon$ or $\sup_{\boldsymbol{x}} |f_1(\boldsymbol{x}) - \rho(\boldsymbol{A})f_2(\boldsymbol{x})| < \epsilon$.*

For simplicity in notation we do not use $\cdot$ and $\odot$ symbols in the next sections. Also in this paper we use $g$ and $h$ to denote group elements and $f$ and $k$ to denote functions.

## 1.2 GROUP CONVOLUTIONAL NEURAL NETWORKS ARCHITECTURE

One conventional method to build group convolutional neural networks is to apply isotropic convolutions for Equation (1). An isotropic $\mathbb{R}^d$ convolution layer maps between planar signals $\mathbb{L}_2(\mathbb{R}^d)$ with $\mathcal{K}$ a planar correlation given by

$$(\mathcal{K}f)(\boldsymbol{y}) = \int_{\mathbb{R}^d} \frac{1}{|\det h|} k(\boldsymbol{x} - \boldsymbol{y}) f(\boldsymbol{x}) \mathrm{d}\boldsymbol{x},$$

and in which $k$ satisfies

$$\text{for all } h \in H : \quad k(\boldsymbol{x}) = \frac{1}{|\det h|} k\left(h^{-1}\boldsymbol{x}\right). \tag{2}$$

Applying isotropic convolutions is limiting because they are constrained by the shape of the kernels. One approach to overcome this limitation, is to lift the signals to the group $G$. Lifting of the input signal, not only addresses the constraints of kernels as noted by (Bekkers, 2019) but also offers advantages in enhancing the performance of image processing, as highlighted in the work by (Smets et al., 2023). When we apply lifting we must look for stabilizer $\text{Stab}_G$ when $G$ acts on $G$. In this case we have

$$H = \text{Stab}_G(g) = \{x \in G | xg = g\} = e.$$

As a result, Equation (2) is fulfilled for all kernels, and there are no longer any limitations imposed on the choice of kernels.

**Definition 10** (Lifting layer ($\mathcal{X} = \mathbb{R}^d$, $\mathcal{Y} = G$)). *Let $G = (\boldsymbol{x}, h)$, where $\boldsymbol{x} \in \mathbb{R}^2$ and $h \in \text{GL}_2(\mathbb{R})$. Also let $k : \mathbb{R}^2 \to \mathbb{R}$ be a compact supported distribution. A lifting layer maps from $\mathbb{L}_2(\mathbb{R}^d)$ to $\mathbb{L}_2(G)$ on the group $G$. A lifting correlation is given by*

$$(\mathcal{K}f)(g) = \int_{\mathbb{R}^d} \frac{1}{|\det h|} k\left(g^{-1}\tilde{\boldsymbol{x}}\right) f(\tilde{\boldsymbol{x}}) \mathrm{d}\tilde{\boldsymbol{x}}.$$

**Example 4** (Lifting for Kronecker delta kernel). *Let*

$$k = \delta(\boldsymbol{x}, \mathbf{0}_{d \times d}) = \begin{cases} 1 & \text{if } \boldsymbol{x} = \mathbf{0} \in \mathbb{R}^d; \\ 0 & \text{otherwise.} \end{cases}$$

*Then for $g = (\boldsymbol{x}, h)$ we have*

$$g^{-1}\tilde{\boldsymbol{x}} = h^{-1}\tilde{\boldsymbol{x}} - \boldsymbol{x}.$$

*Therefore,*

$$k(g^{-1}\tilde{\boldsymbol{x}}) = \delta(h^{-1}\tilde{\boldsymbol{x}} - \boldsymbol{x}, \mathbf{0}_{d \times d}) = \begin{cases} 1 & \text{if } \tilde{\boldsymbol{x}} = h\boldsymbol{x}; \\ 0 & \text{otherwise.} \end{cases}$$

*This results that the lifting layer is as the below*

$$(\mathcal{K}f)(g) = \frac{f(h\boldsymbol{x})}{|\det h|},$$

*which matches with $\rho(g)f$ when $g$ belongs to special linear group $\text{SL}(d)$.*

We also need to discuss this fact that the lifting layer integral exists. A function $f$ on $\mathbb{R}$ is called locally integrable if $f$ is integrable on every bounded interval $[a, b]$ for $a < b$ in $\mathbb{R}$. If $k \in C_c^\infty(\mathbb{R})$ and $f$ is locally integrable, then

$$(f * k)(y) = \int_{-\infty}^{\infty} f(t)k(y - t)dt,$$

exists and is infinitely differentiable on $\mathbb{R}$. First of all the input $f$ is usually a picture and therefore the function $f$ is bounded. On the other hand the value of lifted functions on cosets is equal to that of $f$. Therefore the lifted function is bounded as well. We further know that the kernel is locally supported, which results the integrability. After lifting layer we will apply convolution layer which is defined as follows.

**Definition 11** (Group convolution layer ($\mathcal{X} = \mathcal{Y} = G$)). *A group convolution layer maps between $G$-feature maps in $\mathbb{L}_2(G)$. A group convolution is given by*

$$(f * k)(h) = \int_G f(h)k\left(h^{-1}g\right) \, d\lambda(g),$$

*where $g \in G$ and $\lambda$ is a Haar measure.*

We finally need another layer to again maps to feature maps in $\mathbb{L}_2(\mathbb{R}^d)$, which can be used to imply smoothness of the output.

**Definition 12** (Projection layer). *A projection layer maps between $G$-feature maps in $\mathbb{L}_2(G)$ back to planar feature maps in $\mathbb{L}_2(\mathbb{R}^d)$*

$$(\mathcal{K}f)(\boldsymbol{x}) = \int_{\tilde{H}} f(\boldsymbol{x}, \tilde{h}) d\tilde{h}. \tag{3}$$

## 2 MAIN RESULT

This section reviews the main result of this paper. In the way we explore affine invariant spaces and investigate the convolution integration over $G_2$.

### 2.1 PROBLEM STATEMENT

Our goal is to study invariance in affine transformations in continuous-domain convolutional neural networks. An affine transformation basically combines linear transformations and translations. Affine transformations are denoted as follows

$$G_2 = \left\{ [\boldsymbol{x}, \boldsymbol{A}] : \boldsymbol{x} \in \mathbb{R}^2, \boldsymbol{A} \in \mathrm{GL}_2(\mathbb{R}) \right\},$$

where

$$[\boldsymbol{x}, \boldsymbol{A}] : \boldsymbol{z} \mapsto \boldsymbol{x} + \boldsymbol{A}\boldsymbol{z}.$$

The identity is $[\boldsymbol{0}, \boldsymbol{I}]$, and, therefore, for all $\boldsymbol{B} \in \mathrm{GL}_2(\mathbb{R})$ we have $[\boldsymbol{y}, \boldsymbol{B}]^{-1} = [-\boldsymbol{B}^{-1}\boldsymbol{y}, \boldsymbol{B}^{-1}]$.

The affine transformation is important as we may face affine type distortions due proximity of the camera with respect to the object. For example, this type of affine distortion could manifest in remote sensing images, as well as in camera imagery which can include various perspective distortions (**?**). It is important to note that in an affine transformation, parallel lines in the original image continue to remain parallel in the transformed image. However, the transformation can introduce distortion in the angles between lines.

This paper explores the use of convolutional neural networks in handling affine transformations, focusing specifically on cases where the transformation matrix $\boldsymbol{A}$ belongs to the general linear group $\mathrm{GL}_2(\mathbb{R})$. We diverge from the use of isometric convolutions, opting instead for the application of the lifting-projection method, which we elucidate comprehensively. While prior investigations have focused on compact groups such as SO(2), it is important to highlight that the $\mathrm{GL}_2(\mathbb{R})$ group does not fall under the category of compact groups. As a result, we are unable to apply the Clebsch-Gordan theory to this scenario. The Clebsch-Gordan theory typically addresses criteria for categorizing kernels. In contrast to the traditional approach used to establish the equivalence of two input signals, which relies on solving a challenging optimization problem on the intricate Lie group $G_2$ when dealing with transformations, we opted for a different criterion. Our alternative method focuses on analyzing the convolution of the lifted forms of the signals $f_1$ and $f_2$ for achieving $G_2$ invariance. We also need to introduce an extra step that encompasses performing convolutions on $G_2$ and address the unique challenges associated with this, including techniques for handling integrations over $G_2$. The next theorem asserts that, lifting does not change the affine invariance of input signals

**Theorem 1.** *Let $f_1, f_2 : \mathbb{R}^2 \to \mathbb{R}$ are input signals and let there exists an $g \in G_2$ so that $\sup |(f_1 - \rho(g^{-1})f_2| < \epsilon$ then*

$$\sup |(\mathcal{K}f_1)(g) - \rho(g^{-1})(\mathcal{K}f_2)(g)| < \epsilon \|k\|_1^{\mathbb{R}^2}.$$

*Proof.* We know that $\int_{\mathbb{R}^2} \frac{k(g^{-1}\boldsymbol{x})f(\boldsymbol{x})}{|\det h|} d\boldsymbol{x} = \int_{\mathbb{R}^2} k(\boldsymbol{x})f(g\boldsymbol{x})d\boldsymbol{x}$, then we have

$$\sup_{g'}\left| \left( (\mathcal{K}f_1) - \rho\left(g^{-1}\right)(\mathcal{K}f_1) \right)(g') \right|$$

$$= \sup_{g'}\left| \int_{\mathbb{R}^2} (k(\boldsymbol{x})f_1(g'\boldsymbol{x})d\boldsymbol{x} - k(\boldsymbol{x})f_1(gg'\boldsymbol{x})d\boldsymbol{x} \right|$$

$$\leq \sup_{g'} \int_{\mathbb{R}^2} |k(\boldsymbol{x})||f_1(g'\boldsymbol{x}) - f_1(gg'\boldsymbol{x})|,$$

by setting $g'\boldsymbol{x} = \boldsymbol{y}$ for the last term in above we have

$$\sup_{g'} \int_{\mathbb{R}^2} |k(\boldsymbol{x})||f_1(g'\boldsymbol{x}) - f_1(gg'\boldsymbol{x})| \leq \epsilon \int_{\mathbb{R}^2} |k(\boldsymbol{x})|d\boldsymbol{x} = \epsilon \|k\|_1^{\mathbb{R}^2}.$$

□

Now we provide the main theorems of this paper. The first main theorem states that the invariance of two input signals is related to convolution of lifting of those signals.

**Theorem 2.** *Let* $(\mathcal{K}f_1), (\mathcal{K}f_2) : G_2 \to \mathbb{R}$ *be the lifting of* $f_1, f_2 : \mathbb{R}^2 \to \mathbb{R}$ *and let there exists an* $\tilde{h} \in G_2$ *so that* $\|(\mathcal{K}f_1)(g) - \rho(\tilde{h})(\mathcal{K}f_2)(g)\|_{\sup} < \epsilon$ *then*

$$\|(\mathcal{K}f_1) * k - \rho(\tilde{h})(\mathcal{K}f_2) * k\|_{\sup}^{G_2} < \epsilon\|k\|_1^{G_2}$$

*holds for every kernel* $k$ *and vice-versa. Where* $\|f\|_1^{G_2} = \int_{G_2} |f| d\mu_{G_2}$.

*Proof.* We have

$$\|(\mathcal{K}f_1) * k - \rho(\tilde{h})(\mathcal{K}f_2) * k\|_{\sup}^{G_2} = \sup \left| \int_{G_2} (\mathcal{K}f_1)(g)k(h^{-1}(g)) - \rho(\tilde{h})(\mathcal{K}f_2)(g)k(h^{-1}g)d\mu_{G_2}(g) \right|$$

$$\leq \sup \int_{G_2} \left| (\mathcal{K}f_1)(g)k(h^{-1}(h)) - \rho(\tilde{h})(\mathcal{K}f_2)(g)k(h^{-1}g) \right| d\mu_{G_2}(g)$$

$$\leq \sup \int_{G_2} \left| (\mathcal{K}f_1)(g) - \rho(\tilde{h})(\mathcal{K}f_2)(g) \right| \left| k(h^{-1}(g)) \right| d\mu_{G_2}(g)$$

$$\leq \epsilon\|k\|_1^{G_2}.$$

The second part of the theorem results by selecting $k = \delta(g - h')$. $\square$

The aforementioned finding indicates that to assess the equivalence of two signals, it is necessary to perform a convolutional integration across $G_2$. We investigate this problem in the next section. Now we provide the below theorem which states that the function $c_{(\mathcal{K},k)} : C(\mathbb{R}^2, \mathbb{R}) \to \mathbb{R}$ defined by $\int_{G_2} (\mathcal{K}f) * k \, d\mu_{G_2}(g)$ can be used for characterization of invariant affine functions.

**Theorem 3.** *if* $(\mathcal{K}f_1), (\mathcal{K}f_2) : G_2 \to \mathbb{R}$ *are lifting of input signals and there exists a* $\tilde{h} \in G_2$ *such that* $\|(\mathcal{K}f_1) - \rho(\tilde{h})(\mathcal{K}f_2)\|_1^{G_2} < \epsilon$. *Then we have*

$$\left| \int_{G_2} \big( (\mathcal{K}f_1) * k - (\mathcal{K}f_2) * k \big)(h) d\mu_{G_2}(h) \right| \leq \epsilon\|k\|_1^{G_2}$$

*Proof.* We know that

$$\left| \int_{G_2} \big( (\mathcal{K}f_1) * k - (\mathcal{K}f_2) * k \big)(h) d\mu_{G_2}(h) \right| =$$

$$\left| \int_{G_2} \int_{G_2} \big( (\mathcal{K}f_1)(g)k(h^{-1}g) \big) d\mu_{G_2}(g) d\mu_{G_2}(h) - \int_{G_2} \int_{G_2} \big( (\mathcal{K}f_2)(g)k(h^{-1}g) \big) d\mu_{G_2}(g) d\mu_{G_2}(h) \right|.$$

Then for the second term in the above equation we have and replacing $g$ with $\tilde{h}g$ we have

$$\int_{G_2} \int_{G_2} \big( (\mathcal{K}f_2)(g)k(h^{-1}g) \big) d\mu_{G_2}(g) d\mu_{G_2}(h)$$

$$= \int_{G_2} \int_{G_2} \big( (\mathcal{K}f_2)(\tilde{h}^{-1}g)k(h^{-1}\tilde{h}^{-1}g) \big) d\mu_{G_2}(g) d\mu_{G_2}(h)$$

$$= \int_{G_2} \int_{G_2} \big( (\mathcal{K}f_2)(\tilde{h}^{-1}g)k((\tilde{h}h)^{-1}g) \big) d\mu_{G_2}(g) d\mu_{G_2}(h),$$

if we set

$$f(h) = \int_{G_2} \big( (\mathcal{K}f_2)(\tilde{h}^{-1}g)k((\tilde{h}h)^{-1}g) \big) d\mu_{G_2}(g),$$

then

$$\int_{G_2} \int_{G_2} \big( (\mathcal{K}f_2)(\tilde{h}^{-1}g)k((\tilde{h}h)^{-1}g) \big) d\mu_{G_2}(g) d\mu_{G_2}(h)$$

$$= \int_{G_2} f(h) d\mu_{G_2}(h) = \int_{G_2} f(\tilde{h}h) d\mu_{G_2}(h)$$

$$= \int_{G_2} \int_{G_2} \big( (\mathcal{K}f_2)(\tilde{h}^{-1}g)k(h^{-1}g) \big) d\mu_{G_2}(g) d\mu_{G_2}(h),$$

therefore,

$$\left| \int_{G_2} \int_{G_2} \left( (\mathcal{K}f_1)(g)k(h^{-1}g) \right) d\mu_{G_2}(g)d\mu_{G_2}(h) - \int_{G_2} \int_{G_2} \left( (\mathcal{K}f_2)(g)k(h^{-1}g) \right) d\mu_{G_2}(g)d\mu_{G_2}(h) \right| =$$

$$\left| \int_{G_2} \int_{G_2} \left( (\mathcal{K}f_1)(g)k(h^{-1}g) \right) d\mu_{G_2}(g)d\mu_{G_2}(h) - \int_{G_2} \int_{G_2} \left( (\mathcal{K}f_2)(\tilde{h}^{-1}g)k(h^{-1}g) \right) d\mu_{G_2}(g)d\mu_{G_2}(h) \right| =$$

$$\left| \int_{G_2} \int_{G_2} \left( (\mathcal{K}f_1)(g) - (\mathcal{K}f_2)(\tilde{h}^{-1}g) \right) k(h^{-1}g)d\mu_{G_2}(g)d\mu_{G_2}(h) \right| = \left| \int_{G_2} \left( (\mathcal{K}f_1) - (\mathcal{K}f_2) \circ \tilde{h}^{-1} \right) * k \, d\mu_{G_2}(h) \right|$$

$$\leq \int_{G_2} \left| \left( (\mathcal{K}f_1) - (\mathcal{K}f_2) \circ \tilde{h}^{-1} \right) * k \right| d\mu_{G_2}(h) = \left\| \left( (\mathcal{K}f_1) - (\mathcal{K}f_2) \circ \tilde{h}^{-1} \right) * k \right\|_1^{G_2} \leq \epsilon \|k\|_1^{G_2}.$$

$\square$

## 2.2 CONVOLUTION COMPUTATION

Before illustrating how to compute the convolution over the group $G_2$, we remark some ingredients which is essential to compute the convolution over $G_2$. We finally show that the convolution over $G_2$ can be computed through Fourier transform and integration over real valued space.

In our study, we adopt a straightforward approach to calculate the $G_2$-invariant convolution for a broader kernel, which can be formulated as follows:

$$\int_{G_2} f([\boldsymbol{x}, \boldsymbol{A}])k([\boldsymbol{y}, \boldsymbol{B}]^{-1}[\boldsymbol{x}, \boldsymbol{A}])d\mu_{G_2}. \tag{4}$$

Using the Stone–Weierstrass theorem, in the setup of continuous functions with respect to sup-norm, $C(G_2, \mathbb{R}) = C(\mathrm{GL}_2(\mathbb{R}) \ltimes \mathbb{R}^2, \mathbb{R})$, which asserts that summation of separable functions are dense in $C(G_2, \mathbb{R})$, we reduce the kernel sets to functions of the below form

$$k(\boldsymbol{y}, \boldsymbol{A}) = \sum_{i=1}^{M} k_{1_i}(\boldsymbol{y})k_{2_i}(\boldsymbol{A}).$$

This reduction help us to benefit Fourier transforms to simplify some parts of our calculations. We use QR parametrization of $\mathrm{GL}_2(\mathbb{R})$ which aids us in utilizing numerical approaches, for example are introduced in (Eshkuvatov et al., 2013). Now, we illustrate the outcomes presented in (Schindler, 1993; Milad & Taylor, 2023), which are pertinent to our calculations. Let

$$K_0 = \left\{ \begin{pmatrix} s & -t \\ t & s \end{pmatrix} : s, t \in \mathbb{R}, s^2 + t^2 > 0 \right\},$$

and

$$H_{(1,0)} = \left\{ \begin{pmatrix} 1 & 0 \\ u & v \end{pmatrix} : u, v \in \mathbb{R}, v \neq 0 \right\}.$$

It is shown that $\mathrm{GL}_2(\mathbb{R}) = K_0 H_{(1,0)}, K_0 \cap H_{(1,0)} = \boldsymbol{I}$, where $\boldsymbol{I}$ denotes the identity matrix, and $(\boldsymbol{M}, \boldsymbol{C}) \to \boldsymbol{MC}$ is a homeomorphism of $K_0 \times H_{(1,0)}$ with $\mathrm{GL}_2(\mathbb{R})$. From (Milad & Taylor, 2023) we have

$$\int_{G_n} f \, d\mu_{G_n} = \int_{\mathrm{GL}_n(\mathbb{R})} \int_{\mathbb{R}^n} f[\boldsymbol{x}, \boldsymbol{A}] \frac{d\boldsymbol{x} d\mu_{\mathrm{GL}_n(\mathbb{R})}(\boldsymbol{A})}{|\det(\boldsymbol{A})|}, \text{ for all } f \in C_c(G_n), \tag{5}$$

where $C_c(G)$ denotes the space of continuous $\mathbb{C}$-valued functions of compact support on $G$. For any integrable function $f$ on $\mathrm{GL}_2(\mathbb{R})$, the Haar integral on $\mathrm{GL}_2(\mathbb{R})$ can be expressed as

$$\int_{\mathrm{GL}_2(\mathbb{R})} f \, d\mu_{\mathrm{GL}_2(\mathbb{R})} = \int_{K_0} \int_{H_{(1,0)}} f(\boldsymbol{MC})|\det(\boldsymbol{C})|d\mu_{H_{(1,0)}}d\mu_{K_0}. \tag{6}$$

The map $[u,v] \to \begin{pmatrix} 1 & 0 \\ u & v \end{pmatrix}$ is an isomorphism of the group $G_1 = \mathbb{R} \rtimes \mathbb{R}^*$ with $H_{(1,0)}$. When $n = 1$, $\mathrm{GL}_1(\mathbb{R})$ can be identified with $\mathbb{R}^*$ and $G_1$ identified with $\mathbb{R} \rtimes \mathbb{R}$. We recall that $\int_{\mathbb{R}^*} f d\mu_{\mathbb{R}^*} = \int_{\mathbb{R}} f(b) \frac{db}{|b|}$, where the integral on the right hand side is the Lebesgue integral on $\mathbb{R}$, and

$$\int_{G_1} f \, d\mu_{G_1} = \int_{\mathbb{R}} \int_{\mathbb{R}} f[y,b] \frac{dydb}{b^2}. \tag{7}$$

## 2.3 INTEGRAL OVER $G_2$

A difficult aspect in the implementation of group convolutional neural networks involves performing convolutions across the group. This segment addresses this particular challenge by delving into the problem, which we will break down into the more manageable tasks of calculating Fourier transforms and conducting integrations in real-valued space. We have the below theorem for the integration over $G_2$

**Theorem 4.** *Let* $\boldsymbol{A} = \begin{pmatrix} a & b \\ c & d \end{pmatrix} \in \mathrm{GL}_2(\mathbb{R})$ *and let the kernel is separable meaning that* $k(\boldsymbol{x}, \boldsymbol{A}) = k_1(\boldsymbol{x}) k_2(\boldsymbol{A})$ *and consider the one to one transform between $H$ and $H^*$ so that* $H^*(s, t, u, v, \boldsymbol{B}, \boldsymbol{y}) := H_{f,k}(a, b, c, d, \boldsymbol{B}, \boldsymbol{y})$*, where $a = s - ut$, $c = t + us$, $b = -t/v$, and $d = s/v$, then we have*

$$\int_{G_2} f([\boldsymbol{x}, \boldsymbol{A}]) k([\boldsymbol{y}, \boldsymbol{B}]^{-1}[\boldsymbol{x}, \boldsymbol{A}]) d\mu_{G_2} = \int_{\mathbb{R}} \int_{\mathbb{R}} \int_{\mathbb{R}} \int_{\mathbb{R}} H^*(s, t, u, v, \boldsymbol{B}, \boldsymbol{y}) \frac{dudv}{|v|} \frac{dsdt}{s^2 + t^2}.$$

*where*

$$H_{f,k}(\boldsymbol{A}, \boldsymbol{B}, \boldsymbol{y}) = \frac{k_2(\boldsymbol{A}\boldsymbol{B}^{-1})}{|\det(\boldsymbol{A})||\det(\boldsymbol{B}^{-1})|} \mathcal{F}^{-1}\Big(F(\boldsymbol{u}) K_1(\boldsymbol{B}^\top \boldsymbol{u})\Big).$$

The proof of this theorem is discussed in the appendix. Applying this result we can use the numerical methods in (Eshkuvatov et al., 2013) to compute the former integral as it has singularity in $s = 0, t = 0$. Note that we can write $K_0$ as $\mathbb{R}^+ \rtimes \mathrm{SO}(1)$ where

$$\begin{pmatrix} s & -t \\ t & s \end{pmatrix} = (s^2 + t^2) \times \begin{pmatrix} r\cos\theta & -r\sin\theta \\ r\sin\theta & r\cos\theta \end{pmatrix}.$$

The final step that necessitates computation is the integration within the projection layer. In the context of our affine transformation, the stabilizer is specifically $\mathrm{GL}_2(\mathbb{R})$. We refrain from delving into the intricacies of this process, as it bears resemblance to the earlier scenario.

## 3 CONCLUSION

This research explores how continuous-domain convolutional neural networks operate within affine spaces formed by the generalized linear group $\mathrm{GL}_2(\mathbb{R})$. Affine transformations combine linear transformations and translations. These transformations are ubiquitous because they describe distortions that occurs, for example, when a camera is close to the object being photographed. This study goes further by examining affine-invariant spaces across the entire spectrum of invertible matrix space. Unlike the conventional method for determining the invariance of two input signals, denoted as $f_1$ and $f_2$, under a specialized transformation involves solving a complex optimization problem on the intricate Lie group $G_2$, we adopted an alternative criterion, specifically examining the convolution of the lifted versions of $f_1$ and $f_2$ for $G_2$ invariance. We also applied an additional step, which involves computing convolutions over $G_2$ and addressing the associated challenges, including methods for performing integrations over $G_2$. We explore invariance in convolutional neural networks, and extend its scope beyond the limitations of isometric groups in Euclidean space.

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

## 4 APPENDIX

PROOF OF THEOREM 4

*Proof.* We know that

$$
\int_{G_2} f([\boldsymbol{x}, \boldsymbol{A}])k([\boldsymbol{y}, \boldsymbol{B}]^{-1}[\boldsymbol{x}, \boldsymbol{A}])d\mu_{G_2}
$$
$$
= \int_{G_2} f([\boldsymbol{x}, \boldsymbol{A}])k\big(\boldsymbol{B}^{-1}\boldsymbol{x} - \boldsymbol{B}^{-1}\boldsymbol{y}, \boldsymbol{A}\boldsymbol{B}^{-1}\big)d\mu_{G_2}.
$$

Employing (5) we have

$$
\int_{G_2} f([\boldsymbol{x}, \boldsymbol{A}])k(\boldsymbol{B}^{-1}\boldsymbol{x} - \boldsymbol{B}^{-1}\boldsymbol{y}, \boldsymbol{A}\boldsymbol{B}^{-1})d\mu_{G_2}
$$
$$
= \int_{\mathrm{GL}_2} \int_{\mathbb{R}^2} f[\boldsymbol{x}, \boldsymbol{A}]k(\boldsymbol{B}^{-1}\boldsymbol{x} - \boldsymbol{B}^{-1}\boldsymbol{y}, \boldsymbol{A}\boldsymbol{B}^{-1})\frac{dx_1 dx_2}{|\det(\boldsymbol{A})|}d\mu_{\mathrm{GL}_2},
$$

we also set

$$
H_{f,k}(\boldsymbol{A}, \boldsymbol{B}, \boldsymbol{y}) := \int_{\mathbb{R}^2} f[\boldsymbol{x}, \boldsymbol{A}]k(\boldsymbol{B}^{-1}\boldsymbol{x} - \boldsymbol{B}^{-1}\boldsymbol{y}, \boldsymbol{A}\boldsymbol{B}^{-1})\frac{dx_1 dx_2}{|\det(\boldsymbol{A})|}. \tag{8}
$$

From separability property of kernel we have $k(\boldsymbol{x}, \boldsymbol{A}) = k_1(\boldsymbol{x})k_2(\boldsymbol{A})$. As a result

$$
\begin{aligned}
H_{f,k}(\boldsymbol{A}, \boldsymbol{B}, \boldsymbol{y}) &= \int_{\mathbb{R}^2} f[\boldsymbol{x}, \boldsymbol{A}]k_1(\boldsymbol{B}^{-1}\boldsymbol{x} - \boldsymbol{B}^{-1}\boldsymbol{y})k_2(\boldsymbol{A}\boldsymbol{B}^{-1})\frac{dx_1 dx_2}{|\det(\boldsymbol{A})|} \\
&= \frac{k_2(\boldsymbol{A}\boldsymbol{B}^{-1})}{|\det(\boldsymbol{A})|} \int_{\mathbb{R}^2} f[\boldsymbol{x}, A]k_1(\boldsymbol{B}^{-1}\boldsymbol{x} - \boldsymbol{B}^{-1}\boldsymbol{y})dx_1 dx_2 \\
&= \frac{k_2(\boldsymbol{A}\boldsymbol{B}^{-1})}{|\det(\boldsymbol{A})|}\Big(f * (k_1 \circ \boldsymbol{B}^{-1})\Big) \\
&= \frac{k_2(\boldsymbol{A}\boldsymbol{B}^{-1})}{|\det(\boldsymbol{A})|}\mathcal{F}^{-1}\Big(\mathcal{F}(f)\mathcal{F}(k_1 \circ \boldsymbol{B}^{-1})\Big),
\end{aligned} \tag{9}
$$

where $\mathcal{F}(\cdot)$ denotes the Fourier transform. The next step is to find an explicit form for the Fourier transform. We can apply the result from (Bracewell et al., 1993). Assume that $\mathcal{F}(k_1) = K_1(\boldsymbol{u})$ and $\mathcal{F}(f) = F(\boldsymbol{u})$ then we have

$$H_{f,k}(\boldsymbol{A}, \boldsymbol{B}, \boldsymbol{y}) = \frac{k_2(\boldsymbol{A}\boldsymbol{B}^{-1})}{|\det(\boldsymbol{A})||\det(\boldsymbol{B}^{-1})|} \mathcal{F}^{-1}\Big(F(\boldsymbol{u})k_1(\boldsymbol{B}^\top \boldsymbol{u})\Big).$$

Now we use decomposition of $\mathrm{GL}_2(\mathbb{R})$ as $K_0 \ltimes H(1,0)$ in (Schindler, 1993; Milad & Taylor, 2023).

**Proposition 1** (Proposition 5.1 of (Milad & Taylor, 2023)). *If $\boldsymbol{A} = \begin{pmatrix} a & b \\ c & d \end{pmatrix} \in \mathrm{GL}_2(\mathbb{R})$, then $\boldsymbol{A}$ can be uniquely decomposed as the product $\boldsymbol{A} = \boldsymbol{M_A} \boldsymbol{C_A}$ with $\boldsymbol{M_A} \in K_0$ and $\boldsymbol{C_A} \in H_{(1,0)}$. In fact*

$$\boldsymbol{M_A} = \begin{pmatrix} s & -t \\ t & s \end{pmatrix}, \text{ with } s = \frac{d(ad - bc)}{b^2 + d^2}, t = \frac{-b(ad - bc)}{b^2 + d^2},$$

*and*

$$\boldsymbol{C_A} = \begin{pmatrix} 1 & 0 \\ u & v \end{pmatrix}, \text{ with } u = \frac{cd + ab}{(ad - bc)}, v = \frac{b^2 + d^2}{(ad - bc)}.$$

*This factorization leads to a parallel factorization of $G_2$.*

Consider the one to one transform between $H$ and $H^*$ so that $H^*(s, t, u, v, \boldsymbol{B}, \boldsymbol{y}) := H_{f,k}(a, b, c, d, \boldsymbol{B}, \boldsymbol{y})$, where $a = s - ut$, $c = t + us$, $b = -t/v$, and $d = s/v$. Employing the above proposition and Equation (6) we can write

$$H'(\boldsymbol{B}, \boldsymbol{y}) = \int_{\mathrm{GL}_2} H_{f,k}(\boldsymbol{A}, \boldsymbol{B}, \boldsymbol{y}) d\mu_{\mathrm{GL}_2}$$

$$= \int_{\mathrm{GL}_2} H^*(s(a, b, c, d), t(a, b, c, d), u(a, b, c, d), v(a, b, c, d), \boldsymbol{B}, \boldsymbol{y}) d\mu_{\mathrm{GL}_2}.$$

Therefore, we obtain

$$\int_{\mathrm{GL}_2} H_{f,k}(\boldsymbol{A}, \boldsymbol{B}, \boldsymbol{y}) d\mu_{\mathrm{GL}_2} = \int_{K_0} \int_{H_{(1,0)}} H^*(s, t, u, v, \boldsymbol{B}, \boldsymbol{y}) |v| d\mu_{H_{(1,0)}} d\mu_{K_0},$$

as $\det(\boldsymbol{C_A}) = |v|$. Then we define

$$H^*(s, t, \boldsymbol{B}, \boldsymbol{y}) = \int_{H_{(1,0)}} H^*(s, t, u, v, \boldsymbol{B}, \boldsymbol{y}) \det(\boldsymbol{C_A}) d\mu_{H_{(1,0)}}(u, v)$$

$$= \int_{G_1} H^*(s, t, u, v, \boldsymbol{B}, \boldsymbol{y}) \det(\boldsymbol{C_A}) d\mu_{G_1}(u, v)$$

$$= \int_{\mathbb{R}} \int_{\mathbb{R}} H^*(s, t, u, v, \boldsymbol{B}, \boldsymbol{y}) \frac{dudv}{|v|}.$$

The next step is to compute integration of $H^*(s, t, \boldsymbol{B}, \boldsymbol{y})$ over $K_0$, which is equal to

$$\int_{\mathbb{R}} \int_{\mathbb{R}} H^*(s, t, \boldsymbol{B}, \boldsymbol{y}) \frac{dsdt}{s^2 + t^2}. \tag{10}$$

$\square$

