# OpenReview forum: "AFFINE INVARIANCE IN CONTINUOUS-DOMAIN CONVOLUTIONAL NEURAL NETWORKS"
_ICLR.cc/2024/Conference — ICLR 2024 Conference Withdrawn Submission_

### Official Review · Reviewer_Ts2r · 2023-10-29

**Soundness:** 2 fair
**Presentation:** 2 fair
**Contribution:** 3 good
**Rating:** 3
**Confidence:** 3

**Summary:**

This paper presents a theoretical finding regarding affine invariance in group convolutional neural networks. Namely, the authors:
- Present background material regarding group theory and convolutional neural networks
- Study affine invariance for transforms generated by the full GL2 group, and not for a special case such as SO2
- Show that affine invariance can be measured on lifted signals through convolution on the group
- Discuss the computation of the convolution, and integration, on GL2

**Strengths:**

The findings are interesting to the geometric deep learning community, the authors provide detailed proofs of the theorems, and relaxing the results to the more general GL2 group is a good theoretical improvement.

**Weaknesses:**

As it is, the paper reads closer to a work in progress than to a publishable version.

The paper needs to be re-organised to present the motivation and problem statement, as well as all notations, at the start and not in Section 2.1 (for the problem statement), or scattered throughout Section 1 for notations (e.g. "Also in this paper we use g and h to denote group elements and f and k to denote functions." page 4).

Some of the background material is presented in a sloppy way, e.g.:
- "Definition 2 (Lie groups). Special case of groups are Lie groups, which are symmetries of Riemannian manifolds." this is not the definition of a Lie group
- "Definition 6 (Coset). Let H ⊂ G be a subgroup of G. Then gH denotes a coset given by gH= {g·h|h∈H}" - notations are not defined, and this is the definition of left cosets
- I think the 1 / |det h| in the equation of the planar correlation is redundant with that in Equation 2

All equations should be numbered.

Some of the background material seems presented in the wrong order, e.g.:
- "To address this, we require the concept of group action and group representations. Nevertheless, frequently, our attention is predominantly directed towards linear group actions operating on vector spaces, and these actions are termed representations." (page 4)
- The definition of cosets (that should be rephrased as left cosets) could be presented at the beginning before introducing normal groups

There are issues with references and the presentation of various concepts, e.g.:
- The authors mention multiple times that a major weakness of existing methods is that they rely on solving complex optimization problems over G2, but do not provide any references
- Page 6 - the part about Clebsch-Gordan theory needs references and its relevance to the topic should be introduced
- Page 6 - the reference regarding perspective distortions is missing
- Page 4 "Moreover from (Bekkers, 2019) we know that, if X be a homogeneous space of G. Then X can be identified with G/H with H = StabG (x0) for any x0 ∈ X. Finally we have ε-Affine invariance definition." this is a known result used in Appendix A of Bekkers, 2019 - a better reference would be one of the many textbooks on group theory

All in all, significant improvements are required to present the work in a cohesive way.

Another weakness of this paper is the absence of implementation or experimental results showing whether the theoretical advantages translate to actual group convolutional networks - though I understand the focus is on the theoretical findings.

**Questions:**

"The final step that necessitates computation is the integration within the projection layer. In the context of our affine transformation, the stabilizer is specifically GL2(R). We refrain from delving into the intricacies of this process, as it bears resemblance to the earlier scenario." page 9

Why not include the derivations in the appendix?

Page 6:
"It is important to note that in an affine transformation, parallel lines in the original image continue to remain parallel in the transformed image. However, the transformation can introduce distortion in the angles between lines."
this sounds contradictory, could the authors clarify?

Have the authors implemented the method numerically?

Can their approach be generalized to affine equivariance?

---

### Official Review · Reviewer_WzuJ · 2023-10-31

**Soundness:** 2 fair
**Presentation:** 1 poor
**Contribution:** 2 fair
**Rating:** 3
**Confidence:** 1

**Summary:**

This research studies affine invariance on continuous-domain convolutional neural networks,  focus on the full structure of affine transforms generated by the generalized linear group and introduce a new criterion to assess the similarity of two input signals under affine transformations.

**Strengths:**

The authors introduce a new criterion to assess the similarity of two input signals under affine transformations and analyze the convolution of lifted signals and compute the corresponding integration over $G_2$ which is of some significance.

**Weaknesses:**

1.	The writing of this paper is crude and the layout is poor. Besides, there are some typos, such as ? in the section 2.1 which makes the citing not clear.
2.	The motivation is not convincing and the novelty is not strong.
3.	The authors should conduct experiments to demonstrate the effectiveness of the proposed method, instead of only presenting theoretical results such as Theorems in section 2.

**Questions:**

Please refer to the weakness part.

---

### Official Review · Reviewer_WvUd · 2023-11-03

**Soundness:** 2 fair
**Presentation:** 1 poor
**Contribution:** 3 good
**Rating:** 3
**Confidence:** 4

**Summary:**

The paper is theoretical (no experiments) and describes two things. Firstly, it shows that if two signals are equal (up to some epsilon error) through an affine transformation, then representations obtained by group convolutional neural networks are also similar up to some (scaled) epsilon. This follows from the equivariance principle (theorems 1-3). Secondly, the paper shows how group convolutions for affine groups can be efficiently computed through 1) seperable factorization of the kernel $k(x,A)$ into $k_1(x)k_2(A)$ and 2) making use of the fact that integration over the GL2 part of the affine transformation (over A) can be further split via a QR decomposition. This result is summarized in Theorem 4.

**Strengths:**

* The paper presents an approach which in theory should be of practical value.

**Weaknesses:**

1. Often times, the paper is not very clear. Some definitions are not very precise and often the notation is sloppy (def 2 what is a symmetry of a Riemannian manifold, in def 3 alpha is introduced but not used, example 3 contains a type, equation above (2) uses symbol h, but the expression should not depend on h and moreover h is not defined yet, etc.)
2. The overall structure of the paper is hard to follow. What precisely is the goal of the paper? The paper loosely talks about the problem of matching of functions f1 and f2 without making concrete how this happens. It only defines the notion epsilon-affine invariance but not how we determine when to functions are epsilon invariant. Moreover, the paper on several occasions says that it it wants to avoid solving complex optimization problems (please be more precise on what kind of problems) but does not explain how the paper does it instead. I suppose the motivation is that one can learn representations of functions through affine equivariant mappings, and then measure similarity between those learned representations, but I can only guess.
3. Theorems 1 to 3 all seem to be intimately related and could probably be merged, in particular 1 and 2 seem almost identical. Could these not be merged by stating that affine invariance is maintained for any equivariant mapping $\Phi$ (with the property $\Phi[\rho(g) f] = \rho'(g) \Phi[f]$, with $\rho$ and $\rho'$ representations for the in and output resp.). An intuitive explanation up front would also help; why are these theorems relevant?
4. Section 2.2 then seems to be the most important result but feels a bit rushed and the essence of it is hard to grasp (I hope my summary above is correct).
5. The objective of the paper seems to be efficiency, but there is no proof of principle on the actual usefulness of the approach. I.e., I imagine that splitting the convolution over 4 integrals, including forward and backward Fourier transforms and change of variables might not be more efficient than applying out of the box group convolutions.
6. Example 4 is incorrect and is the result from an earlier typo.

Smaller comments:
1. The abstract uses $G_2$ as a symbol, but it is unclear what it represents. (The affine Lie group $R^2 \rtimes GL_2(\mathbb{R}^2)$, I imagine). I don't think it is ever defined.
2. There are 3 papers which might be relevant based on the notion of separability of group convolutions. [Knigge et al. 2022] Uses separability convolutions to achieve equivariance to scale-rotation-translation transforms.  [Chen et al. 2021] uses separability for efficient implementations for SE(3) equivariance. [Mironenco and Forre 2023] have almost exactly the same results as this paper (decomposition of the GL group), and also  builds on the works from the group of Taylor, so it seems. This last reference however appeared after the ICLR submission deadline so I of course do not expect you to have known about this paper, it could however a helpful reference in improving the narrative of the paper.

[knigge et al] Knigge, D. M., Romero, D. W., & Bekkers, E. J. (2022, June). Exploiting redundancy: Separable group convolutional networks on lie groups. In International Conference on Machine Learning (pp. 11359-11386). PMLR.

[chen et al] Chen, H., Liu, S., Chen, W., Li, H., Hill, R.: Equivariant point network
for 3d point cloud analysis, 14514–14523 (2021)

[Mironenco and Forre] Mironenco, M., & Forré, P. (2023). Lie Group Decompositions for Equivariant Neural Networks. arXiv preprint arXiv:2310.11366.

3. The contributions again say rather then focussing on complex optimization problems, we study invariance through convolution integrals. I do not follow this logic. Simply studying invariance obviously does not require an optimization problem. What precisely is the alternative to the complex optimization problem? This is unclear to me. I do understand the implications of the theorems, but no where in the paper do I learn how these results are to be used in practice.

4. What is the reason of writing presenting the kernel transformation (above definition 1)? I think part of the story is missing that tells that under and equivariance constraint this operator reduces to a group convolution? I am guessing this is the story because the paper so far closely follows the paper Bekkers 2019 as well as his lecture series on Youtube (might be a useful reference to add? https://uvagedl.github.io)

5. Example 3 contains a typo. It should be $f(R^{-1}_\theta (y - x)$, the parentheses were missing. This mistake is later on used in Example 4. The resulting kernel operator in example 4 is therefore also false.

6. In definition 2 it is unclear what is meant with symmetries of a Riemannian manifold, it seems like an odd way to define a Lie group.

7. In definition 4, alpha is defined but not used. Instead $\odot$ is used but not defined.

8. In section 1.2 the isotropic kernel convolution is defined, but this definition should not have the 1/|det h| in it, because the left-hand side does not depend on $h$, and moreover, $|\det h| = 1$ if $H=SO(d)$ (which is also not defined).

9. Example 4 is incorrect as $g^{-1} \tilde{x} = R^{-1}(\tilde{x} - x)$ and the resulting $\mathcal{K}$ should be $(\mathcal{K} f)(x,h)=f(x)/|\det h|$.

10. In section 2.1 when $G_2$ finally gets defined, you should also define the group product. Without a definition of a group product the definition of the group is not complete. Currently only the action on $\mathbb{R}^d$ is given. I know this is a bit nit-picky because the full group product merely involves the product of the GL2 part (A.B).

11. The short discussion on the Clebsch-Gordan theory is a bit odd. It is not necessarily used for "categorizing kernels", rather I would say it is a framework for working with irreducible representations and tensor products relative to basis for vector spaces that transform via the irreps. And then, I do not see the connection to the subsequent sentence "In contrast to the traditional approach used to establish...". Some logic here is missing.

12. In theorem 1, there are two types of $g$... One is used as input coordinates to the feature maps, the other is used to parametrize the transformation.

**Questions:**

So comments above.

---

### Official Review · Reviewer_taA6 · 2023-11-03

**Soundness:** 2 fair
**Presentation:** 2 fair
**Contribution:** 2 fair
**Rating:** 3
**Confidence:** 3

**Summary:**

The authors consider the construction of equivariant neural network
architectures that process signals defined on $\mathbb{R}^2$ (say, images),
which should be invariant/equivariant to affine transformations of domain. This
application is important for processing images of the physical world, where
affine transformations of an image correspond under 'local' conditions to
viewing the same scene content from a different camera angle, and hence preserve
local semantic features of the input image. The authors develop mathematically a
methodology that lifts convolutional filters over the images themselves to
convolutional integrals over the affine group $\mathrm{GL}(2) \rtimes
\mathbb{R}^2$, then projects down to obtain a filtered image, in order to
construct these affine-equivariant filters.

**Strengths:**

- The authors consider the important problem of affine invariance in equivariant
  neural networks for computing with visual data. This specific problem is of
  great interest among equivariant networks that operate on visual data, and it
  is often outside the realm of applicability of studies that focus only on
  $G$-equivariance with respect to Lie group actions on the signal itself
  (rather than on its domain).

- The work presents mathematically-rigorous analysis.

**Weaknesses:**

- The paper would benefit strongly from an empirical component -- even verifying
  the computational prescription encompassed by the main result, Theorem 4, in
  the setting of a toy example. This is a significant aspect of the paper's
  motivation, and it therefore feels like a significant omission to not have any
  such practical verification.

- The paper would benefit from substantial polishing of the presentation. Here
  are three concrete areas:
  - General motivation and connection to prior work. The problem the authors
    consider is very important, in my view (as I wrote in the strengths
    section). However, the authors do not present a very robust argument for the
    fundamental importance of this problem (mentioning only "such distortions
    arise in photos when the camera is close to the subject being captured" in
    the intro and conclusion, and applications to CAPTCHA). The authors could
    make reference to works on 3D correspondence (say, textbooks -- Hartley and
    Zisserman; Košecká, Sastry, Soatto, and Ma) that emphasize
    the ubiquity and fundamental nature (both from a "practical" perspective and
    mathematical perspective) of affine invariance in visual data.
    They could also discuss more precisely how existing methods have struggled
    to treat this framework, with in particular a more precise discussion of
    related works that have similar motivations (such as works on steerable
    networks which process similarity transformations or scale, by Weiler, Cohen, and others, which also involve a treatment of the noncompact setting).
  - Precision of presentation. The organization of the paper renders it
    challenging to read. Section 1.1 presents three pages of background
    material, including definitions and notation that will not be necessary to
    parse the main results of the paper. Key concepts, such as convolution
    integrals on abstract LCH spaces $\mathcal{X}$ and on topological groups $G$
    with Haar measure, as well as the associated "liftings" that play a key role
    in the theory, are not emphasized as clearly as a consequence of this
    clutter, and the notation is correspondingly confusing (e.g., note
    $\mathcal{K}{\boldsymbol{w}}$ in the display at top of page 3, the later
    notation $\mathcal{K}$ for convolution kernels in equation (2), induced
    liftings $\mathcal{K}$ defined in Definition 10, and projection layers in
    Definition 12, all with identical notation). Measures of integration are
    often hard to parse (note $d\lambda$ in Definition 11, $d\mu_{\mathcal{X}}$
    in the display at top of page 3, and later $d\mu_{G_2}$ in the main
    methodology; which of these measures is (left-? bi-?) invariant Haar measure
    and which are simply general measures often needs to be inferred from
    context. This will make the paper laborious to parse for a non-expert, and
    limit the ability of readers to use the authors' methodology for
    computational improvements -- doubly important given that such experiments
    are not given in the paper.
  - General polish of writing. The paper would benefit from a robust
    proofreading for typos, grammatical errors, and notational imprecisions, as
    well as general polish for tone. Examples of imprecise language include "Our
    focus are affine spaced formed by the generalized linear group ..." in the
    intro (the affine group involves translations too; similar issue below with
    "...expressed as $\mathrm{SO}(n)$"); Definition 2 (this definition of a Lie
    group cannot be parsed without knowing what a "symmetry" and a "Riemannian
    manifold" is); the notation $G_2$ for the affine group on $\mathbb{R}^2$
    used throughout the paper (why not use a standard notation for this set?);
    "First of all, the input $f$ is usually a picture..." on page 5; a broken
    (LaTeX) reference on page 6; suprema throughout section 2 which have
    'type errors' (see for example Theorem 1's statement, with $g$). These are
    only a small selection.

**Questions:**

- Equation (4) does not include a term for the determinant of either of the
  matrices $\boldsymbol{A}$ or $\boldsymbol{B}$, which seemed to play an
  important role in the earlier theory, eg Definition 10 and Theorem 1. Can this
  omission be justified?

---

### Meta-Review · Area_Chair_C9cw · 2023-12-06

**Metareview:**

This theory paper considers the construction of $GL(2)$-invariant networks for signals defined over the domain $\mathbb{R}^2$.  The authors describe lifting, convolution, and projection layers over the group $GL(2)$ and prove these layers preserve affine invariance and show how to compute convolutions over $GL(2)$ efficiently using separable convolution and QR decomposition.  Reviewers were unanimous that the paper is not ready for publication pointing out many specific issues with the paper presentation and structure.  Most reviewers also felt the paper was incomplete as a pure theory paper and should include some empirical evaluation.  Ts2r and Wzuj also commented the motivation should be better explained.  That said, taA6 noted the work was rigorous and taA6 and WvUd concurred the work was on an interesting problem and could be theoretically useful.

 The consensus is that this paper needs substantial revision before publication and could benefit from experiments.  The reviewers have provided the authors with substantial specific feedback on how to improve the paper. I suggest they revise their work and submit to a new venue.

**Justification For Why Not Higher Score:**

- paper presentation
- no experiments
- reviewer consensus

**Justification For Why Not Lower Score:**

N/A

---

### Decision · Program_Chairs · 2024-01-16

Reject